# NR-DARTS: NODE REWIRING FOR DIFFERENTIABLE ARCHITECTURES WITH ADAPTIVE SE-FUSION

## ABSTRACT

Efficient model design is critical for deployment on edge and embedded hardware where compute, latency, and energy budgets dominate feasibility. To make efficiency a property of the architecture under these budgets, Neural Architecture Search (NAS) and post-hoc pruning methods are widely used to discover task specific backbones that meet deployment constraints. However, conventional channel or operation level pruning is ill suited to NAS cells since local saliency proxies are unreliable under multi branch interactions and weight sharing, and fine grained removals break cell wise dimensional coupling and trigger cascading realignments. Thus, we propose NR-DARTS, Node Rewiring for Differentiable Architectures with Adaptive SE Fusion, which deletes low importance intermediate nodes scored by learnable gates. Then, the proposed method rewires their predecessors directly to each successor, and compensates at the successor input via a learned linear aggregation followed by channel wise SE recalibration. By preserving cell structure and feature dimensional consistency, our method avoids misalignment issues common in fine grained pruning and achieves reliable performance. On CIFAR-10 dataset, NR-DARTS reduces FLOPs by 27.3% from 338.94M to 246.41M while maintaining accuracy at 93.81% versus 94.07% for the DARTS baseline and it outperforms channel and operation level pruning under matched budgets. Ablation studies further show that adaptive SE fusion improves accuracy at similar FLOPs compared to fixed summation and explain the effectiveness of the compensation mechanism.

## 1 INTRODUCTION

Resource-constrained edge and embedded hardware impose strict compute, latency, and energy budgets, elevating efficiency to a primary design objective (Sze et al., 2017). Accordingly, efficiency is evaluated by reducing MACs/FLOPs as well as latency and energy, rather than by parameter count alone. In such settings, task-specific backbones tailored to the target application and hardware budget are preferable to task-agnostic designs (Tan et al., 2019). Under these constraints, efficiency-aware architecture discovery is naturally framed as a search problem, for which Neural Architecture Search (NAS) provides a principled mechanism to obtain candidate architectures under deployment budgets (Elsken et al., 2019).

In practice, NAS often evaluates candidate architectures using proxy mechanisms such as weight sharing and partial training. Among differentiable NAS methods, Differentiable Architecture Search (DARTS) instantiates this setting by relaxing discrete operation choices to a continuous parameterization and performing gradient-based bilevel optimization over shared supernet weights and architecture parameters (Liu et al., 2018a). This proxy-based evaluation introduces an estimation bias that favors candidates adept at rapid proxy loss reduction rather than those with the best fully trained performance (Zela et al., 2019; Ye et al., 2022). Therefore, a post-hoc pruning phase, such as structured pruning, is essential to refine the discovered topology and improve its accuracy-efficiency trade-off.

Post-hoc pruning is widely used to compress trained networks and has also been applied to NAS-discovered architectures, typically by removing channels or individual operations (Cheng et al., 2017). However, this strategy faces two key challenges. First, pruning criteria based on local importance proxies, such as weight magnitude or gradient sensitivity, are unreliable in the multi-branch structures typical of NAS (Chu et al., 2021). Second, fine-grained removal changes channel counts

and breaks the dimensional coupling within each searched cell, leading to cascading realignments that weaken the architecture and reduce accuracy (Fang et al., 2023).

To address these limitations, we propose *NR-DARTS*, a post-hoc pruning framework that refines NAS-discovered topologies while preserving their structural consistency. Our approach makes the following key contributions:

- We introduce NR-DARTS, a node-level pruning framework that evaluates the holistic importance of nodes using learnable gates, ensuring reliable pruning decisions.
- Our method preserves NAS cell dimensional consistency by pruning entire nodes, avoiding misalignment issues common in fine-grained approaches.
- Our retraining framework, enhanced with an adaptive SE-fusion module to recover information flow, achieves a 27.3% reduction in FLOPs on CIFAR-10 relative to the DARTS baseline, with accuracy maintained within 0.26 percentage points

## 2 RELATED WORKS

**Neural Architecture Search**   Neural Architecture Search (NAS) has been proposed as a systematic approach to automate network design, with reinforcement learning and evolutionary methods achieving strong performance but suffering from excessive computational costs (Zoph & Le, 2016; Real et al., 2017). Gradient-based methods such as DARTS have greatly reduced the search cost (Liu et al., 2018a). However, these methods still rely on proxy evaluations with weight sharing and partial training. This reliance introduces bias, favoring architectures that quickly reduce proxy loss rather than those that achieve the best final performance (Ye et al., 2022). Various strategies have been proposed to address this issue, including regularization, improved supernet training, and fairness mechanisms (Xu et al., 2019; Chu et al., 2021; Dooley et al., 2023; Jeon et al., 2025). While these methods improve supernet evaluation to some extent, they mainly focus on training-time fairness and ranking correlation. As a result, challenges remain in achieving optimal accuracy–efficiency trade-offs for practical deployment scenarios.

**Neural Network Pruning**   Pruning has long been studied as a model compression technique, initially focusing on unstructured weight pruning to reduce storage cost (Han et al., 2015). However, such approaches often yield limited real speedups on hardware. Structured pruning methods later emerged, removing channels, filters, or even entire layers to achieve practical efficiency gains (Li et al., 2016; He et al., 2017). Despite their effectiveness, these methods were primarily developed for manually designed networks such as ResNet and VGG, and applying them to NAS-discovered architectures remains challenging. The modular and interdependent nature of NAS cells makes fine-grained pruning difficult, as it may break structural consistency and degrade performance. To overcome these limitations, recent efforts have attempted to integrate pruning into the NAS pipeline (Ding et al., 2022; Jiang et al., 2023). While effective, these methods couple the search process tightly with hardware-specific constraints. In contrast, our approach decouples performance-oriented search from efficiency optimization. This decoupling enables a single high-performance model to be flexibly adapted to diverse deployment budgets through post-hoc refinement.

## 3 PROPOSED METHOD

### 3.1 MOTIVATION

Conventional structured pruning methods face inherent difficulties when applied to complex NAS-discovered architectures. This difficulty arises because such fine-grained methods pose fundamental challenges regarding both the reliability of the pruning criteria and the structural consistency of the resulting architecture .

In terms of reliability, conventional structured pruning typically scores component importance using local proxies such as filter magnitude (Li et al., 2016). These metrics are limited because signal strength or parameter size does not guarantee functional relevance (Liu et al., 2018b; Blalock et al., 2020). A large-norm filter can be redundant, whereas a small-norm filter can be critical for accuracy (Molchanov et al., 2019). In NAS-discovered architectures, weight sharing and multi-branch

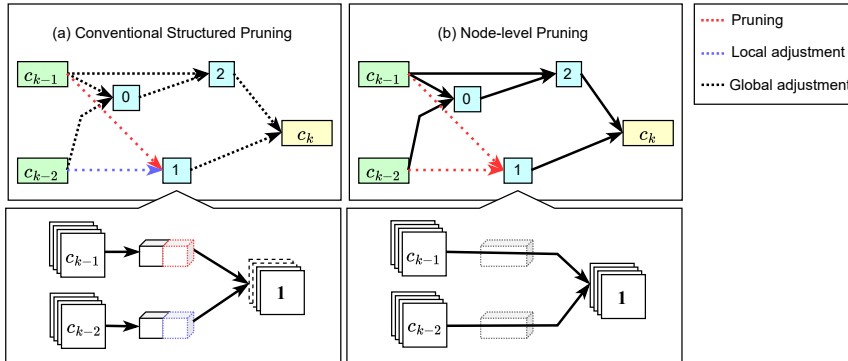

Figure 1: A conceptual comparison of pruning granularities in cells discovered by NAS. (a) Conventional structured pruning operates at a fine-grained level, targeting components within an operation. (b) Node-level pruning operates at a coarser, structural level by removing an entire computational node.

aggregation further distort per-branch estimates, as observed in FairNAS (Chu et al., 2021), making local proxies especially unreliable. Consequently, pruning in multi-branch NAS cells becomes challenging because such biases obscure the true contribution of each branch.

Furthermore, structural consistency is fragile under conventional structured pruning, because changing channel counts forces alignment across nodes and cells (He et al., 2017). As illustrated in Figure 1(a), removing a single channel first requires a local adjustment in the directly connected node by removing the corresponding channel position to preserve input–output compatibility. Because NAS cells are designed with a fixed channel width per node and all incoming edges to a node must share the same channel dimension (Tan et al., 2019), such local adjustments do not remain isolated. They propagate across the cell as a global adjustment, requiring the same channel removal across operations to maintain structural consistency. This cascading change not only forces modifications in nodes unrelated to the original decision but also can amplify feature-dimension misalignment beyond the intended scope, weakening the structure discovered by NAS (Fang et al., 2023).

To address the structural and reliability issues discussed above, we raise the pruning granularity from individual filters or operations to the node level (Figure 1(b)). This shift in granularity addresses both challenges simultaneously. For reliability, we attach learnable gates to the outputs of nodes during a dedicated evaluation phase, enabling a more stable assessment of the functional contribution of each node beyond biased local proxies. For structural consistency, node-level pruning naturally preserves the dimensional compatibility required within NAS cells.

Simply deleting a node, however, would disrupt information flow to its successors. Therefore, pruning at the node level must be accompanied by rewiring, in which the predecessors of the deleted node are directly connected to its successors. This step maintains connectivity while avoiding the cascading alignment issues observed in channel-level pruning. Building on this principle, we propose NR-DARTS, a node-level pruning and rewiring framework that combines structure-aware consistency with reliable importance estimation.

## 3.2 PROPOSED METHOD

Our proposed NR-DARTS is a post-hoc framework designed to achieve structural lightweighting and robust retraining of architectures discovered by NAS. The framework consists of three distinct stages to ensure both efficiency and performance, as illustrated in Figure 2.

The first stage of our framework is the Pruning Search, where the objective is to quantitatively estimate the structural importance of each intermediate node within a discovered DARTS architecture(Liu et al., 2018a). To achieve this, we introduce a set of learnable gates, denoted by $\gamma_k$, which are applied to the output of each node, as visually depicted in Figure 2(a). Applying the gates

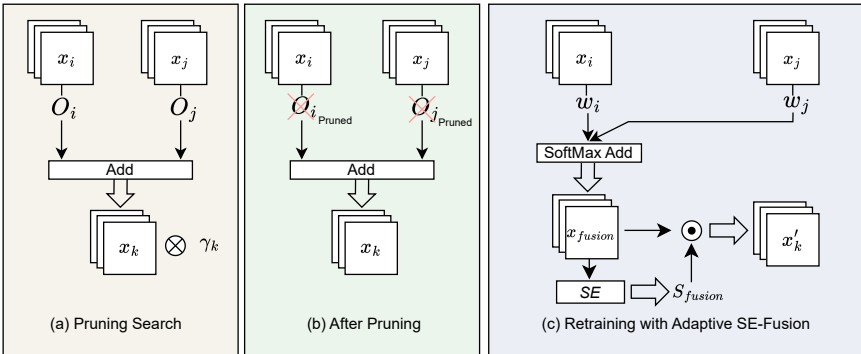

Figure 2: An overview of the NR-DARTS framework, illustrating the transformation of a node across three stages. (a) Structural importance of each node is estimated using learnable gates $\gamma_k$ during the search process. (b) Operations within nodes with low importance scores are pruned. (c) The proposed Adaptive SE-Fusion module recalibrates the altered information flow, and the pruned network is retrained from scratch.

directly to node outputs, rather than estimating importance from local proxies, yields importance scores that more accurately reflect the functional contribution of a node (Gao et al., 2018). The gates are applied exclusively to the intermediate nodes within the normal cells. This selective application strategy is motivated by the architectural characteristics of the network. Normal cells are repeated far more frequently than reduction cells(Liu et al., 2018a). Their cumulative contribution dominates the overall computational cost of the network. As a result, pruning the intermediate nodes in normal cells offers the greatest potential for model compression. In contrast, reduction cells perform the critical function of spatial down-sampling. Altering their structure may compromise multi-scale feature representation and destabilize gradient propagation (Mao et al., 2017). Preserving reduction cells is therefore essential for maintaining the hierarchical integrity of the network.

This gating mechanism is mathematically formulated as:

$$
x_k = \gamma_k \left( \sum_{i \in \mathcal{P}(k)} O_{i \to k}(x_i) \right), \quad \gamma_k = \begin{cases} \tilde{\gamma}_k, & k \in \mathcal{N}_{\text{normal}}, \\ 1, & k \in \mathcal{N}_{\text{reduction}}, \end{cases} \quad \text{with } \tilde{\gamma}_k \in \mathbb{R}. \quad (1)
$$

Defined in (1), the output of an intermediate node $k$ is modulated by a node-level gate $\gamma_k$. Specifically, the outputs from its predecessor nodes $x_i$ ($i \in \mathcal{P}(k)$) are first transformed by their corresponding operators $O_{i \to k}$ and summed. This aggregated result is then scaled by the single gate $\gamma_k$, which uniformly controls the contribution of node $k$. For nodes in normal cells ($\mathcal{N}$normal), $\gamma_k$ is initialized to 1 and optimized as a learnable parameter together with the operator weights $O$, enabling the estimation of node level structural importance. For nodes in reduction cells ($\mathcal{N}$reduction), $\gamma_k$ is fixed to 1, and their structure is preserved without modification. Thus, the optimized gate values in normal cells directly quantify node importance and serve as pruning indicators in the subsequent stage.

After the Pruning Search, the second stage prunes the architecture based on the learned importance scores, as shown in Figure 2(b). Nodes are ranked by their gate values $\gamma_k$ in ascending order to determine their relative structural importance, and a predefined ratio of the lowest-scoring nodes is pruned. This pruning ratio is treated as a hyperparameter set according to the overall network configuration. Each pruned node is simplified into a parameter-free summation block by removing all internal operations ($O_{i \to k}$), and the resulting lightweight architecture is retrained from scratch to optimally adapt to the reduced structure.

In the final stage of our framework, the pruned nodes remain as simple summation blocks, which can result in information loss due to the removed operations. Following the principle of feature map approximation used in ThiNet (Luo et al., 2017), this loss can be alleviated by retraining the network to restore the original feature representations. To achieve this, we introduce the Adaptive SE-Fusion

module. This module adaptively weights the contributions of predecessor feature maps and performs channel-wise recalibration, effectively compensating for the missing information.

This process can be mathematically formalized as:

$$x_{\text{fusion}} = \sum_{i \in \mathcal{P}(k)} w_i' x_i, \quad w_i' = \frac{\exp(\tilde{w}_i)}{\sum_j \exp(\tilde{w}_j)}$$

$$S_{\text{fusion}} = \sigma \left( W_2 \, \delta \left( W_1 \, \text{GAP}(x_{\text{fusion}}) \right) \right)$$

$$x_k' = x_{\text{fusion}} \odot S_{\text{fusion}} \tag{2}$$

$x_{\text{fusion}}$ is the intermediate feature map created by the adaptive aggregation of predecessor node outputs. Empirically, we observe that softmax-normalized weights $w_i'$ often result either in a dominant single-branch contribution or in meaningful multi-branch interactions, a phenomenon that has also been observed in prior work (Chu et al., 2021). To address this, our framework employs linear approximation to learn the relative importance of each input path, allowing it to flexibly capture both sparse single-branch dominance and more complex multi-branch interactions. To further refine the linearly fused features, we introduce a non-linear, channel-wise recalibration via a Squeeze-and-Excitation (SE) block (Hu et al., 2018). The SE block produces a channel attention vector $S_{\text{fusion}}$, derived from the global context of $x_{\text{fusion}}$. The refined output $x_k'$ is obtained by applying element-wise multiplication between $x_{\text{fusion}}$ and $S_{\text{fusion}}$. This non-linear recalibration enables the model to dynamically emphasize informative channels while suppressing less relevant ones, resulting in a more context-aware representation.

Through the three stages, the proposed framework constructs a network that efficiently compresses the architecture while preserving essential information.

## 4 EXPERIMENTAL RESULTS

### 4.1 EXPERIMENTAL SETTINGS

All experiments were conducted on the CIFAR-10 dataset, comprising 50,000 training images. The dataset was partitioned into 75% for model training and 25% for validation. To ensure reproducibility, we used random seeds 41, 42, and 43, and report results as the mean and standard deviation across the three runs. Prior to training, images were normalized using per-channel means of 0.491, 0.482, and 0.446, and standard deviations of 0.247, 0.244, and 0.262. Data augmentation consisted of random cropping to a resolution of 32×32 and random horizontal flipping.

Experiments were implemented in PyTorch 2.4.1 with CUDA 11.8, and executed on a workstation equipped with an NVIDIA RTX 4060 GPU and an AMD Ryzen 5 7500F CPU. All hyperparameters followed the official DARTS implementation (Table 4), while the fusion module parameters were optimized independently.

To rigorously evaluate the effectiveness of the proposed model, we conducted a comparative study against reference models derived from the same DARTS backbone, as summarized in Table 5. This setup enables a fair comparison with conventional pruning methods, including magnitude based, gradient based, and BN scale pruning, since all methods are applied to the same architecture. The evaluation is based on MFLOPs to measure computational cost, Top-1 accuracy to assess classification performance, and latency to reflect practical execution efficiency.

### 4.2 COMPARISON RESULTS

Table 1 reports the results of our comparative experiments. We evaluate whether the proposed method achieves a better balance between accuracy, efficiency, and latency than conventional structured pruning. The analysis focuses on maintaining accuracy while reducing FLOPs, and on examining whether these changes consistently improve runtime. The table summarizes the performance of all considered models. Specifically, the best result in each column is highlighted in bold, and the second-best result is underlined.

Table 1: Comparison of pruning methods on architectures discovered by DARTS

| Category | Model | Top-1 Acc (%) | ΔTop-1 (%) | FLOPs (M) | ΔFLOPs (%) | Latency (ms) |
|---|---|---|---|---|---|---|
| Baseline | Base DARTS | $94.07 \pm 0.08$ | — | 338.94 | — | 5.59 |
| Proposed method | NR-DARTS | $\mathbf{93.81 \pm 0.05}$ | **-0.26** | 246.41 | -27.3 | 4.92 |
| Pruning methods | Gradient-based pruning | $93.67 \pm 0.03$ | -0.40 | 248.70 | -26.6 | 5.45 |
| | L1 pruning | $93.36 \pm 0.09$ | -0.71 | 248.70 | -26.6 | 5.23 |
| | L2 pruning | $93.35 \pm 0.05$ | -0.72 | 248.70 | -26.6 | 5.25 |
| | LAMP pruning | $93.35 \pm 0.05$ | -0.72 | 248.70 | -26.6 | 6.49 |
| | FPGM pruning | $93.35 \pm 0.05$ | -0.72 | 248.70 | -26.6 | 5.24 |
| | Random pruning | $93.23 \pm 0.08$ | -0.84 | 248.70 | -26.6 | 5.12 |
| | BN scale pruning | $92.98 \pm 0.20$ | -1.09 | 248.70 | -26.6 | 5.19 |
| | Reduced DARTS | $92.66 \pm 0.71$ | -2.10 | **244.53** | **−27.8** | **3.62** |

The proposed NR-DARTS achieves a Top-1 accuracy of 93.81% while reducing the FLOPs of the Base DARTS model by 27.3% and improving latency from 5.59 ms to 4.92 ms. These results indicate that substantial computational savings can be obtained with only a marginal 0.26% drop in accuracy, demonstrating the effectiveness of the proposed efficient pruning framework. In particular, NR-DARTS shortens latency more effectively than conventional methods, providing a tangible benefit in practical deployment where fast response time is critical.

In contrast, the Reduced DARTS achieves comparable FLOPs reduction and improves latency to 3.62 ms, but its accuracy drops by 1.41%. This suggests that naive depth reduction compromises the hierarchical feature representations that are crucial for high performance. This observation has been empirically validated by early deep architectures such as VGGNet (Simonyan & Zisserman, 2014). By contrast, our method adopts node-level pruning rather than reducing depth. This strategy preserves sufficient architectural depth while discarding only less important nodes. As a result, NR-DARTS achieves competitive latency reduction without severe accuracy degradation, effectively balancing efficiency and representational capacity.

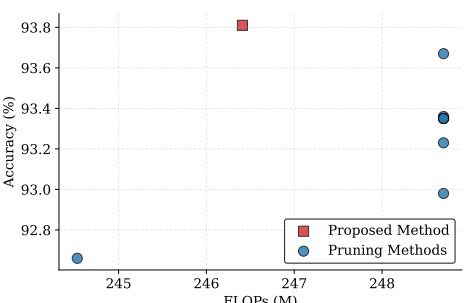

Figure 3: FLOPs vs accuracy comparison between pruning methods and NR-DARTS.

Compared to conventional structural pruning methods adjusted for similar FLOPs, NR-DARTS consistently achieves superior accuracy while also reducing inference latency. In particular, it improves accuracy by up to 0.83 percentage points and reduces latency by 4–24% over conventional methods, whose performance ranges from 92.98–93.67% accuracy and 5.12–6.49 ms latency.

Conventional structural pruning, previously discussed as a source of concern, suffers from structural inconsistency and unreliable performance estimation. Experimental results suggest that these issues may contribute to reduced accuracy and increased or variable latency. In contrast, NR-DARTS performs pruning at the node level, which appears to alleviate structural inconsistency in practice. In addition, it applies gates directly to node outputs, suggesting a more reliable importance evaluation. A comprehensive comparison is further illustrated in the accompanying summary Figure 3.

### 4.3 IN-DEPTH ANALYSIS

In this section, we provide an in-depth analysis of the proposed framework through both quantitative and qualitative evaluations. We first conduct ablation studies to examine the effects of pruning strategies and fusion mechanisms. We then complement these results with qualitative analyses using t-SNE visualizations. Additional qualitative analyses with HiResCAM are provided in the Appendix.

### 4.3.1 ABLATION STUDY

We conduct ablation studies to disentangle the contributions of two key components in our framework. The first analysis focuses on the pruning strategy adopted during the search stage, and the second examines the role of the SE Fusion module in mitigating performance degradation after pruning.

We first analyze the effect of different pruning strategies during the search stage. Our goal is to understand how node selection impacts the accuracy, stability, and computational cost of the final model (FLOPs). Specifically, we compare our proposed low-importance pruning, which removes nodes with minimal contribution, with random pruning as a baseline and high-importance pruning as a negative control. This setup allows us to systematically evaluate whether selectively pruning low-importance nodes preserves model performance while reducing computational cost. After retraining, overall accuracy remains high across all strategies, but differences are observed between them.

As summarized in Table 2 and shown in Figure 4, the Low strategy, which prunes low-importance nodes, achieves the highest mean accuracy (93.81±0.05%) while maintaining relatively high FLOPs (246.41 M). Random pruning results in lower accuracy (93.30 ± 0.19%) and intermediate FLOPs (220.51 M), and pruning high-importance nodes yields the lowest accuracy (93.09 ± 0.13%) and FLOPs (204.06 M). Notably, the Low strategy also exhibits the smallest standard deviation, indicating stable performance across runs.

This comparison clearly highlights the underlying accuracy–FLOPs trade-off. Pruning high-importance nodes severely damages accuracy by eliminating critical components of the computational skeleton of the network. Random pruning, meanwhile, offers no predictable benefit, yielding a suboptimal and unprincipled compromise between metrics. In contrast, the effectiveness of the Low strategy lies in its methodical preservation of this core structure. By selectively targeting only the least influential nodes, it ensures the integrity of the feature hierarchies of the network remains intact. This approach therefore achieves substantial computational savings while minimizing performance degradation, striking an optimal and reliable balance between accuracy and efficiency.

Table 2: Comparison of accuracy and FLOPs across pruning strategies after retraining.

| Strategy | Accuracy (%) | FLOPs (M) |
|----------|--------------|-----------|
| Low      | 93.81 ± 0.05 | 246.41    |
| Random   | 93.30 ± 0.19 | 220.51    |
| High     | 93.09 ± 0.13 | 204.06    |

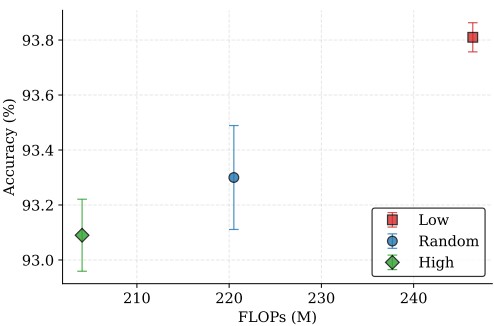

Figure 4: Comparison of accuracy and FLOPs trade-off across pruning strategies; low importance pruning achieves the best balance.

Next, we evaluate the contribution of the SE Fusion module by comparing it against alternative fusion strategies. This analysis highlights how different fusion designs recover information lost during pruning, allowing us to determine which approach best preserves accuracy under comparable computational budgets. Table 3 presents their performance and FLOPs, while Figure 5 provides a visual comparison.

With comparable FLOPs across strategies (240–247M), SE Fusion achieved the highest Top-1 accuracy of 93.81%, with only a 0.26% drop compared to the base DARTS, while reducing FLOPs by 27.3%. This result experimentally validates our design choice. The combination of linear fusion and non-linear channel-wise SE calibration effectively compensates for pruning-induced distribution shifts and mitigates accuracy degradation.

Learnable Fusion replaces channel-wise SE calibration with a single global scaling factor and achieved 93.46%. This suggests that the absence of non-linear channel-level correction limits its ability to capture fine-grained variations. Arithmetic schemes such as Mean, Max, and Add yielded slightly lower accuracies, ranging approximately from 93.34% to 93.42%. This suggests that naive feature aggregation cannot fully correct pruning-induced scale shifts. Multiplicative fusion further amplified these distortions. Accuracy dropped to 92.95%. Finally, the Zero strategy confirmed that the network remains functional without explicit fusion. However, it suffered the largest degradation, with accuracy falling to 92.44%.

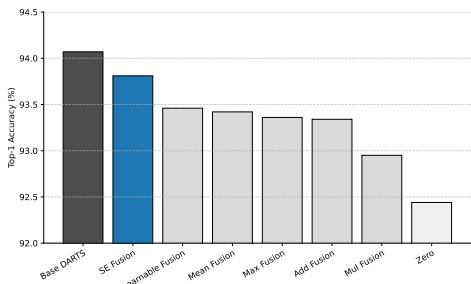

Figure 5: Top-1 accuracy vs. FLOPs for fusion strategies; SE Fusion best preserves accuracy under comparable computational budgets.

Overall, these results demonstrate that SE Fusion provides the most effective balance between preserving accuracy and maintaining computational efficiency. This highlights the importance of combining linear aggregation with non-linear channel-wise calibration.

Table 3: Comparison of Top-1 accuracy and FLOPs for different fusion strategies in NR-DARTS ablation.

| Fusion Strategy | Top-1 Acc (%) | $\Delta$Top-1 (%) | FLOPs (M) | $\Delta$FLOPs (%) |
|---|---|---|---|---|
| Base DARTS | $94.07 \pm 0.08$ | — | 338.94 | — |
| SE Fusion | $93.81 \pm 0.05$ | $-0.26$ | 246.41 | $-27.3$ |
| Learnable Fusion | $93.46 \pm 0.15$ | $-0.61$ | 243.99 | $-28.0$ |
| Mean Fusion | $93.42 \pm 0.12$ | $-0.65$ | 243.47 | $-28.2$ |
| Max Fusion | $93.36 \pm 0.06$ | $-0.71$ | 240.86 | $-28.9$ |
| Add Fusion | $93.34 \pm 0.12$ | $-0.73$ | 242.44 | $-28.4$ |
| Mul Fusion | $92.95 \pm 0.39$ | $-1.12$ | 242.95 | $-28.3$ |
| Zero | $92.44 \pm 0.12$ | $-1.63$ | 242.44 | $-28.4$ |

### 4.3.2 QUALITATIVE ANALYSIS OF LEARNED REPRESENTATIONS

We use t-SNE (t-Distributed Stochastic Neighbor Embedding) to examine feature distributions. All analyses are conducted on the same four models: the original unpruned model (Base DARTS) as the upper-bound reference, Gradient-based Pruning and BN Scale Pruning as baselines, and our proposed NR-DARTS.

We analyze the feature distributions of the four models using t-SNE to understand the global structure of the learned representations. Specifically, the test-set feature vectors are embedded into two dimensions with perplexity=40 and n_iter=1000, which allows us to visually compare how pruning affects the learned representations in each model.

As shown in Figure 6, the Base DARTS model exhibits clearly separated clusters for each class, achieving the highest Silhouette Score of 0.518. NR-DARTS exhibits comparable intra-class cohesion and inter-class separation with a Silhouette Score of 0.510. Conventional pruning methods, in contrast, lead to a clear degradation in the feature space. Gradient-based pruning produces partially overlapping clusters, resulting in a reduced Silhouette Score of 0.476. BN Scale pruning causes an even more pronounced structural collapse, as clusters not only overlap but also lose internal cohesion, sharply lowering the score to 0.407.

These results indicate that NR-DARTS, specifically designed for NAS-discovered architectures, effectively approximates the original discriminative feature representations. Through its incorporation of SE fusion, the method compensates for lost or weakened features due to pruning. This mechanism also mitigates linear and non-linear distortions, maintaining intra-class cohesion and inter-class sep-

aration. In contrast, conventional structured pruning, which is not tailored for NAS architectures, can disrupt the latent feature space and reduce cluster clarity. These findings highlight the importance of pruning methods specialized for NAS-discovered architectures in preserving representational structure.

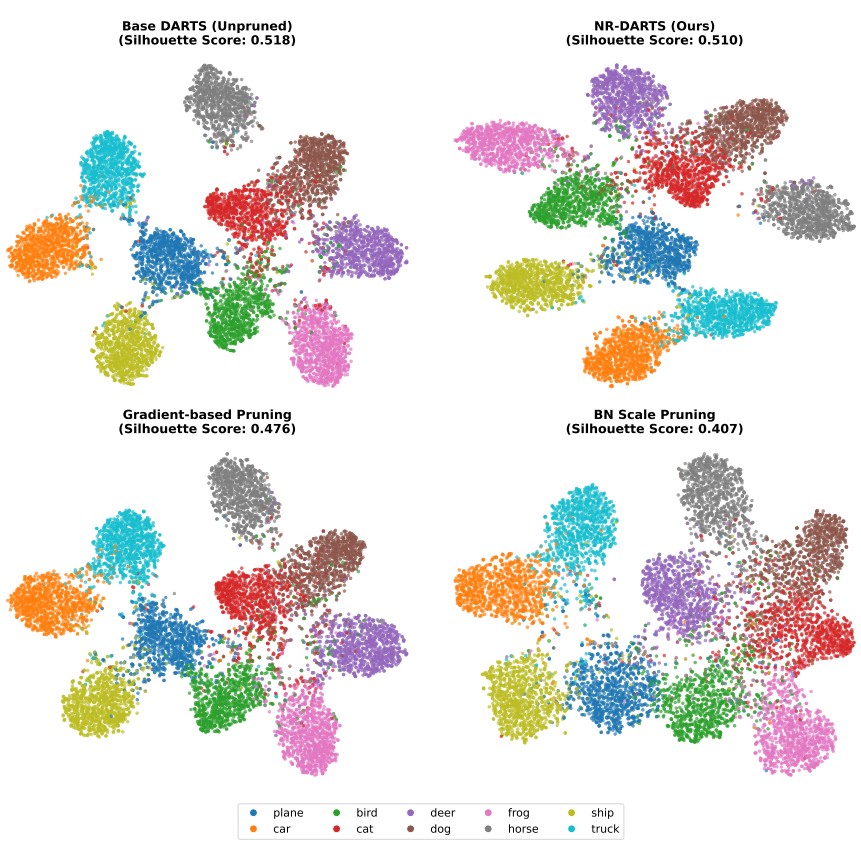

Figure 6: t-SNE visualization of test-set feature embeddings showing cluster separation quality. NR-DARTS (0.510) maintains comparable discriminative structure to Base DARTS (0.518), while conventional pruning methods show reduced cluster cohesion (0.476, 0.407)

## 5 CONCLUSION

In this paper, we addressed the accuracy–efficiency trade-off of NAS-discovered architectures by proposing a node-level pruning approach specialized for discovered structures. Unlike prior channel and operation focused pruning methods, our approach elevated the pruning unit to the node level, reducing structural inconsistencies while using gate-based importance estimation for reliable node evaluation. On CIFAR-10, our proposed NR-DARTS achieved a 27.3% reduction in FLOPs with only a 0.26% drop in accuracy, demonstrating both efficiency and performance preservation. These results suggested that node-level pruning can serve as a complementary strategy to existing NAS compression methods, while also indicating the potential to extend practical applicability. However, our study is limited to small datasets and models, and relies on a separated search and training pipeline. Future work will investigate larger NAS benchmarks, real-world edge deployment, and integrated search-and-train frameworks for end-to-end optimization.

## REPRODUCIBILITY STATEMENT

Details of the experimental setup are in Section 4.1, the complete set of hyperparameters is provided in Appendix A, and an anonymized implementation is available at `https://anonymous.4open.science/r/NR-DARTS-D4B2`.

## LLM USAGE

We used large language models (LLMs) solely to assist in polishing the manuscript. All scientific content, results, and interpretations were written and verified by the authors.

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

# A  APPENDIX

This appendix provides additional details about the experimental setup and comparison baselines used in the main paper. We first summarize the hyperparameter configurations shared across all experiments and then describe the models considered in our comparative study.

## A.1  HYPERPARAMETER SETTINGS

Table 4 lists the hyperparameters used throughout search, training, and fine-tuning. Shared parameters include the optimizer configuration, learning rate schedule, and initialization details that were consistent across all methods. Phase-specific parameters indicate the settings that differ across the three phases: the search phase for optimizing node importance, the train phase for training pruned architectures with SE fusion, and the fine-tuning phase, applied only to baseline models with pruning methods, for retraining them to recover performance.

Table 4: Hyperparameter settings

| Parameter | Value |
|---|---|
| *Shared parameters* | |
| Layers | 8 |
| Batch size | 64 |
| Initial learning rate | 0.025 |
| Optimizer | SGD |
| Momentum | 0.9 |
| Weight decay | $3 \times 10^{-4}$ |
| Drop path probability | 0.3 |
| Initial channels | 16 |
| *Phase-specific parameters* | |
| Epochs (search / train / fine-tune) | 120 / 100 / 10 |
| Fusion configuration | 10-epoch warm-up, LR=0.002, dropout=0.3 |
| Total nodes pruned in network | 15 |

Table 5: Overview of comparison models

| Category | Model | Description |
|---|---|---|
| Baseline | Base DARTS | Standard 8-layer DARTS architecture |
| Proposed Method | **NR-DARTS (Ours)** | **Our method with Node Pruning and SE-Fusion** |
| Pruning Methods | Reduced DARTS | 6-layer variant of DARTS for efficiency |
| | L1 Pruning | Pruning based on L1-Norm of filter weights |
| | L2 Pruning | Pruning based on L2-Norm of filter weights |
| | LAMP Pruning | Layer-wise magnitude-based pruning |
| | FPGM Pruning | Pruning based on the geometric median |
| | BN Scale Pruning | Pruning based on BatchNorm $\gamma$ scaling |
| | Gradient-based Pruning | Pruning based on gradient sensitivity |
| | Random Pruning | Randomly removes filters as a baseline |

## A.2 COMPARISON MODELS

Table 5 provides an overview of all models evaluated in our experiments. The baseline is the standard 8-layer DARTS architecture. Our proposed method, NR-DARTS, augments DARTS with node-level pruning, rewiring, and SE-based fusion. For comparison, we include a reduced-depth variant (Reduced DARTS) as well as a range of structured pruning methods applied to the DARTS backbone, covering magnitude-based pruning (L1, L2, LAMP), geometric median pruning (FPGM), BatchNorm-scale pruning, gradient-based pruning, and a random pruning baseline. All models are trained under the shared hyperparameters described in Table 4, ensuring fairness in the evaluation of accuracy, FLOPs, and runtime.

## A.3 HIRESCAM ANALYSIS OF SPATIAL ATTENTION

We further analyze spatial attention patterns of the same four models by visualizing class activation maps with HiResCAM applied to the final convolutional layer. This evaluation highlights how different pruning and fusion strategies influence the spatial focus of the models, revealing which approaches best preserve critical regions and feature localization under comparable computational budgets.

Figure 7 shows four representative scenarios. In these cases, NR-DARTS exhibits sharper and more focused attention, effectively suppressing background noise while retaining discriminative cues. In the performance improvement cases (Images 1 and 2), the Base DARTS model misclassified the inputs due to poorly localized attention. Its activation maps are diffuse and fail to concentrate on the most discriminative features of the objects, such as the head of the bird or the wings of the plane. In contrast, NR-DARTS exhibits sharp, well-localized attention that precisely highlights these key features, leading to correct predictions with high confidence of 0.99. This demonstrates its ability to recover and emphasize critical cues. For the precise attention distribution case (Image 3), all models correctly predicted the dog. NR-DARTS, however, focused more strongly on the facial features and body contours of the dog, yielding the highest confidence of 1.00. This suggests that the model not only preserves correctness but also enhances the localization of discriminative features. In the knowledge preservation case (Image 4), both Base DARTS and NR-DARTS attended to the salient regions of the ship. Both models achieved identical performance with a confidence of 1.00, confirming that key representational knowledge remains intact. Finally, in the failure case (Image 5), all models misclassified the cat. However, NR-DARTS produced notably lower confidence of 0.36, indicating better uncertainty calibration.

Together, these observations suggest that NR-DARTS can maintain both accuracy and spatial fidelity while providing improved model reliability in the examined cases.

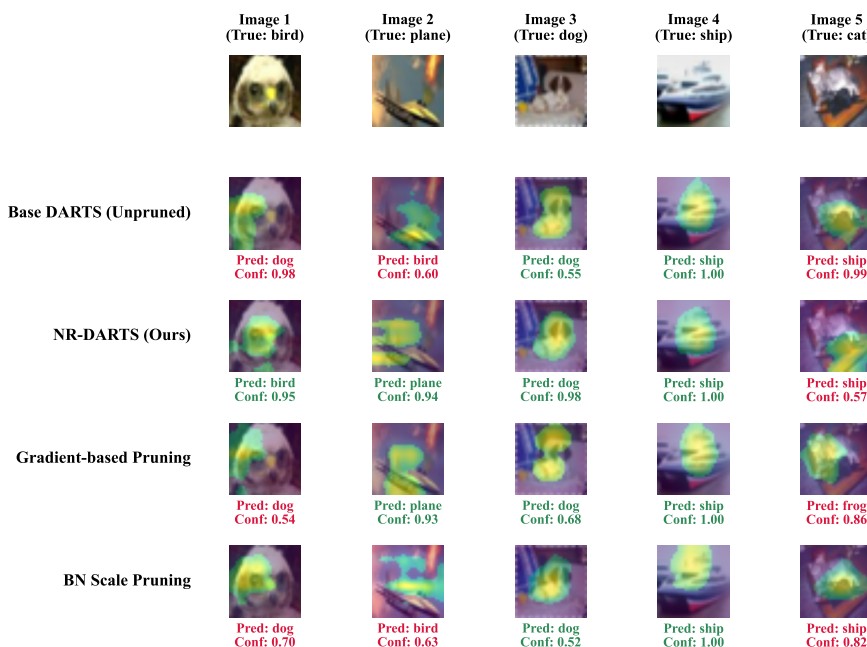

Figure 7: HiResCAM visualizations for five representative cases: NR-DARTS sharpens attention on key features, correcting Base DARTS errors (1–2), refines correct predictions (3–4), and lowers confidence on failures (5).