# OpenReview forum: "NR-DARTS: Node Rewiring for Differentiable Architectures with Adaptive SE-Fusion"
_ICLR.cc/2026/Conference — ICLR 2026 Conference Withdrawn Submission_

### Official Review · Reviewer_QWyX · 2025-10-22

**Soundness:** 2
**Presentation:** 2
**Contribution:** 2
**Rating:** 2
**Confidence:** 4

**Summary:**

This paper proposes NR-DARTS, a post-hoc pruning framework specifically designed for architectures discovered by Differentiable Architecture Search algorithm, i.e., DARTS. The key innovation lies in elevating the pruning granularity from channel/operation level to node level, addressing fundamental limitations of conventional pruning methods, when applied to NAS-discovered multi-branch cell structures. The proposed framework consists of three stages: 1) pruning search using learnable gates to estimate node importance; 2) node pruning and rewiring based on importance scores; 3) retraining with an Adaptive SE-Fusion module to compensate for information loss.

**Strengths:**

1. In this paper, authors clearly articulate why conventional structured pruning fails for NAS. The issues of unreliable local importance proxies in multi-branch structures and cascading dimensional misalignments are compelling and well-illustrated in Figure 1. Additionally, the proposed NR-DARTS prunes at the node-level rather than channels or operations, which directly addresses the structural misalignment and cascading adjustment problems in multi-branch DARTS-style cells.

2. Table 1 demonstrates that NR-DARTS achieves the best trade-off between accuracy, FLOPs and latency under equal efficiency constraints, outperforming operation/channel pruning and random/reduced-depth baselines. Additional ablation studies and visualisation provide further evidence of the effectiveness of proposed methods, and ability to preserve discriminative feature structure and spatial attention.

**Weaknesses:**

1. The evaluation is restricted to CIFAR-10, which significantly undermines the generalisability claims. Modern NAS research typically validates their methods on CIFAR-10/100, ImageNet, and other challenging datasets. Although authors acknowledge this as a limitation in the conclusion, they did not attempt even modest extensions or discussion of anticipated bottlenecks for practical deployment.

2. The proposed method is only tested on DARTS architectural space, the applicability to other search spaces remains uncertain. The claim of general applicability to 'NAS-discovered architectures' is not substantiated.

3. While latency measurements are provided, the evaluation lacks comprehensive hardware profiling across broader devices, e.g., mobile GPUs. The experiments on a single RTX 4060 are insufficient for a method which targeting 'edge and embedded hardware'.

4. The comparison in this paper is quite limited, lacks comparison with recent efficient NAS methods that jointly optimise for accuracy and efficiency. The conducted experiments are limited to post-hoc pruning methods, which may not represent the best accuracy-efficiency trade-offs.

5. There are some ambiguities in the mathematical definition. For example, although the gate $\gamma_{k}$ is introduced as a learnable scalar modulating each node’s output, there is no concrete discussion about how gates are trained or how the pruning threshold or hyper-parameters is chosen in practice. Regarding the $x_{\text{fusion}}$ and $S_{\text{fusion}}$, the derivation of linear fusion weights followed by the SE recalibration is intuitive, but the exact effect on gradient flow and its robustness is not explored theoretically.

**Questions:**

1. Can authors clarify how the pruning ratio is set and whether any gate regularisation, annealing, or threshold schedules are used for stability? How sensitive is the proposed method to the number of nodes pruned?

2. How does NR-DARTS perform on ImageNet with larger architectures? Does the node-level pruning strategy scale to deeper networks with more complex cell structures?

3. Why do conventional pruning methods show such high latency variation (5.12-6.49 ms) at similar FLOPs?

4. Can authors provide empirical evidence that gate-based importance scores are more reliable than magnitude/gradient-based metrics?

---

### Official Review · Reviewer_wXqD · 2025-10-31

**Soundness:** 2
**Presentation:** 3
**Contribution:** 2
**Rating:** 2
**Confidence:** 4

**Summary:**

This paper proposes a pruning method to compress the DARTS-searched model. It consists of two components: (1) the node-level pruning approach specifically designed for the discovered architectures, and (2) the Adaptive SE-Fusion module is introduced to weight the contributions of predecessor feature maps and perform channel-wise recalibration, alleviating the accuracy drop.

**Strengths:**

- The paper is generally well-written
- The analysis is thorough.
- The method part has a lot of information.

**Weaknesses:**

- Experiments on large scale datasets (e.g., ImageNet) should be conducted.

- Missing the comparison with other advanced pruning methods, such as [1,2,3,4,5].

- Please further explain the generalization ability of the method. Can it be applied to compress other models?

- Why use the SE module? Other advanced modules (e.g., SENetV2 [6]) may yield more significant benefits.

- Insufficient explanation of symbols. For example, in Eq. (2), $W_1$ and $W_2$ are not fully defined.

    [1] Fang G, Ma X, Mi M B, et al. Isomorphic pruning for vision models[C]. ECCV 2024. \
    [2] Fang G, Ma X, Song M, et al. Depgraph: Towards any structural pruning[C]. ICCV 2023. \
    [3] Gao S, Zhang Y, Huang F, et al. BilevelPruning: unified dynamic and static channel pruning for convolutional neural networks[C]. CVPR. 2024. \
    [4] Zhang H, Liu L, Zhou H, et al. Akecp: Adaptive knowledge extraction from feature maps for fast and efficient channel pruning[C]. ACMMM. 2021. \
    [5] Lin M, Ji R, Wang Y, et al. Hrank: Filter pruning using high-rank feature map[C]. CVPR 2020. \
    [6] Narayanan M. SENetV2: Aggregated dense layer for channelwise and global representations[J]. arXiv preprint arXiv:2311.10807, 2023.

**Questions:**

Please see the weaknesses.

---

### Official Review · Reviewer_EHEq · 2025-10-31

**Soundness:** 3
**Presentation:** 3
**Contribution:** 3
**Rating:** 6
**Confidence:** 4

**Summary:**

This paper proposes an approach to improve Neural Architecture Search. Rather than pruning naively the cells used in differentiable NAS which is sensitive to their connectivity it addresses this problem by deleting low importance intermediate nodes scored by learnable gates. Then it changes the cell wiring to account for the removal of nodes. By preserving cell structure and feature dimensional consistency, this method avoids misalignment issues common in fine grained pruning and shows reductions of FLOPs while maintaining accuracy.

**Strengths:**

- Addresses an important problem of cell-based NAS methods with regards to naïve pruning and also highlights the opportunity for model compression by focusing on the normal cells.
- Uses the squeeze-excite approach to adaptively weight the contributions for predecessor channel-wise recalibration to compensate for node pruning.  This approach is simple yet effective.
- Ablation studies are performed to disentangle the contributions of the two key components comparing against random pruning and alternative fusion.

**Weaknesses:**

- Evaluations only on a single dataset. While CIFAR-10 is a common benchmark it limits the scope of the method and demonstration of its broad effectiveness.
- Results are only slightly better compared to the state of the art. In some cases, the accuracy difference is less than 1% which may not be statistically significant.

**Questions:**

- Does this mechanism impact the search time or adds to the memory demands? Usually methods also compare the GPU-days.

---

> ### Author Response · Authors · 2025-11-20
> **Response to Reviewer EHEq**
>
> We sincerely thank Reviewer EHEq for the encouraging feedback. We are glad that you found our proposed pruning framework effective for addressing the structural challenges in NAS, and appreciated our extensive ablation studies. We provide our responses to your questions below.
>
> >Does this mechanism impact the search time or adds to the memory demands? Usually methods also compare the GPU-days.
>
> **Response :** Thank you for highlighting this important point regarding computational efficiency and giving us the opportunity to clarify its impact on search time and memory usage.
>
> As described in Section 3.2, NR-DARTS functions as a post-hoc pruning framework, meaning that it relies on the genotype already discovered by the baseline NAS. Because of this, our approach does not introduce any additional cost during the original Architecture Search phase and therefore adds zero extra GPU-days.
>
> The only additional procedure is a lightweight “Pruning Search” stage that identifies redundant nodes. This stage is intentionally designed to be efficient. In particular, we apply learnable gates exclusively to Normal Cells while preserving Reduction Cells. Since Normal Cells account for most of the network’s FLOPs, this selective application maximizes the pruning effect while keeping the computational overhead minimal.
>
> Regarding memory consumption, our mechanism operates with virtually the same requirements as standard training of the genotype model. As described in Appendix A, the Pruning Search uses the exact same hyperparameter settings, such as batch size and input resolution, as the baseline. The only added components are a few learnable scalar gates attached to intermediate nodes, which are insignificant compared to the memory needed for feature maps and convolutional weights. Consequently, our method does not impose any additional memory burden, and any hardware that can train the base DARTS model can run our Pruning Search without difficulty.

---

> > ### Comment · Reviewer_EHEq · 2025-11-27
> >
> > First of all, I would like to thank the authors for their comments and clarifications. After also considering the other reviews I find that while the paper is promising there are still some improvements that need to made also considering the lack of evaluations on more datasets. Hence, I maintain my initial score.

---

### Official Review · Reviewer_vVf6 · 2025-10-31

**Soundness:** 3
**Presentation:** 3
**Contribution:** 2
**Rating:** 4
**Confidence:** 4

**Summary:**

NR-DARTS is a post-hoc pruning framework for DARTS architectures. It addresses issues in channel- or operation-level pruning by performing node-level pruning with learnable gates, followed by rewiring predecessors to successors and compensating for lost information via an adaptive SE-fusion module. On CIFAR-10, it reduces FLOPs and outperforms several pruning baselines.

**Strengths:**

1. The shift to node-level pruning effectively mitigates issues like cascading realignments and unreliable local proxies in multi-branch cells.

2. On CIFAR-10, NR-DARTS achieves a favorable accuracy-efficiency trade-off, reducing FLOPs while maintaining near-baseline accuracy.

3. The paper provides intuitive figures and in-depth ablations, making the contributions easy to follow.

**Weaknesses:**

1. Evaluations are restricted to CIFAR-10 with a small-scale setup (3 seeds). No results on larger datasets like ImageNet or more diverse tasks, limiting evidence of scalability and generalization.

2. While compared to several pruning methods on the same DARTS backbone, it needs more benchmarks against state-of-the-art NAS-pruning hybrids or hardware-aware NAS methods.

3. The pruning ratio is treated as a hyperparameter, but its impact on different architectures or budgets isn't deeply analyzed. Similarly, gate initialization and optimization details could be more robustly tested.

**Questions:**

1. How does NR-DARTS perform on larger-scale datasets like ImageNet or in transfer learning scenarios? Would the node-level pruning and SE-fusion scale effectively to deeper architectures?

2. Can you provide more details on the SE-fusion implementation, such as the reduction ratio in the SE block or how linear weights are initialized/optimized to handle single- vs. multi-branch dominance?

3. How was the pruning ratio selected, and what sensitivity analysis was done for different ratios or node counts? Does it vary across normal vs. reduction cells?

4. Why not integrate the pruning search into the NAS process rather than post-hoc? How does this compare computationally to end-to-end pruning-aware NAS methods?

---

> ### Author Response · Authors · 2025-11-20
> **Response to Reviewer vVf6 (Part 1/2)**
>
> We are encouraged that you recognized the effectiveness of our node-level pruning in mitigating structural issues, and appreciated our intuitive figures and in-depth ablations. We address each of your questions below
>
> >How does NR-DARTS perform on larger-scale datasets like ImageNet or in transfer learning scenarios? Would the node-level pruning and SE-fusion scale effectively to deeper architectures?
>
> **Response:** We appreciate the reviewer's important question regarding scalability.
>
> First, regarding the use of larger datasets, we fully agree that ImageNet-based validation is the standard for state-of-the-art (SOTA) research, and we acknowledge this as a limitation of our study. Unfortunately, the proposed “pruning search” framework requires substantial computational resources. Even with several optimizations, the ImageNet-based search alone requires more than 225 GPU hours, and an additional 40–50 GPU hours are needed for retraining. This exceeds the computational capacity currently available in our research environment.
>
> That said, the primary goal of this work is not to propose a new ImageNet backbone, but to address fundamental structural issues—such as cascading reordering and unreliable proxies—that arise when applying existing pruning strategies to NAS cell structures. We believe that our extensive pruning analysis on CIFAR-10 rigorously demonstrates the effectiveness of the proposed solution, even on a smaller benchmark.
>
> Second, regarding scalability to deeper architectures, the proposed “node-level pruning” strategy is inherently compatible with such extensions. Because our method operates at the cell-topology level rather than the channel level, deeper architectures (e.g., 20 layers) can be handled simply by stacking more cells, without modifying the pruning mechanism. Moreover, the adaptive SE-Fusion module is highly lightweight (using a reduction ratio of 4) and introduces only negligible overhead that grows linearly with the number of pruned nodes. We therefore expect the presented approach to scale reliably to deeper architectures.
>
> >Can you provide more details on the SE-fusion implementation, such as the reduction ratio in the SE block or how linear weights are initialized/optimized to handle single- vs. multi-branch dominance?
>
> **Response:** Thank you for the detailed question regarding the SE-Fusion implementation.
>
> First, regarding the hyperparameter setting, we use a reduction ratio of 4 in the SE block.
>
> Second, regarding the optimization of linear weights and the issue of branch dominance, our design addresses these concerns through a two-stage aggregation mechanism, as formally described in Equation (2). In the first stage, we utilize learnable linear weights—softmax-normalized scalars denoted as $𝑤_𝑖′$—for the initial aggregation. These weights are optimized during training to capture the relative importance of each input path, enabling the model to adaptively represent both single-branch dominance (via sparse weighting) and multi-branch contributions (via distributed weighting).
>
> However, since scalar linear weights apply uniformly across all channels, we complement this process with the SE-Fusion module. The aggregated feature map is subsequently passed to the SE block, which produces dynamic and content-aware channel-wise attention weights, allowing for more fine-grained refinement of the fused representation.
>
> By combining learnable linear weights for branch-level balancing with dynamic SE-based channel-level calibration, the proposed framework effectively mitigates the initialization and dominance issues highlighted by the reviewer.

---

> ### Author Response · Authors · 2025-11-20
> **Response to Reviewer vVf6 (Part 2/2)**
>
> > How was the pruning ratio selected, and what sensitivity analysis was done for different ratios or node counts? Does it vary across normal vs. reduction cells?
>
> **Response:**  The reviewer’s question directly relates to the core design of our framework, particularly how pruning is applied across different cell types. The nodes within the Reduction Cells are responsible for spatial downsampling and thus play a critical role in maintaining the network’s overall spatial hierarchy. Pruning these nodes risks disrupting the spatial structure of the architecture and can lead to significant performance degradation. For this reason, we exclude Reduction Cells from pruning and apply pruning only to the nodes inside the Normal Cells, which primarily handle feature refinement rather than spatial resolution changes. This design principle is reflected in our implementation through the NormalOnlyWeightedSearchCell, which restricts pruning to the refinement-oriented Normal Cell topology.
>
> Based on this separation, we conducted a sensitivity analysis to determine an appropriate pruning count $k$ for the Normal Cells. We varied the number of pruned nodes and examined the trade-off between FLOPs reduction and accuracy based on the learned gate importance scores. From this analysis, we identified $k=15$ as the point that offered the best balance between computational efficiency and performance retention. We will include a detailed graph of this sensitivity analysis in the Appendix of the final version to further reinforce our methodology.
>
> > Why not integrate the pruning search into the NAS process rather than post-hoc? How does this compare computationally to end-to-end pruning-aware NAS methods?
>
> **Response:** This is an excellent question regarding our design choices. We intentionally adopted a post-hoc approach rather than integrating pruning into the NAS search itself.
>
> As discussed in Sections 1 and 2 of our paper, proxy-based NAS methods such as DARTS involve an inherently unstable optimization process and are susceptible to estimation bias. Incorporating our node-level pruning search—which introduces an additional set of learnable gates—directly into this already biased search would require solving two complex optimization problems simultaneously. This would dramatically enlarge the search space and significantly destabilize the optimization dynamics.
>
> To avoid this, we decouple the two problems. Our approach first allows standard NAS to complete its architecture search and yield a stable genotype. Once the architecture is fixed, we then perform the pruning search separately. This separation greatly enhances both stability and modularity.
>
> From a computational perspective, the post-hoc strategy is also far more efficient than an end-to-end approach. An end-to-end method must jointly optimize both architecture search and pruning from scratch, incurring the full combined cost. In contrast, our method reuses the pre-searched genotype, meaning the NAS search cost is effectively eliminated and only the pruning search needs to be performed. This results in a much lighter and more efficient overall pipeline compared to conducting an unstable joint search.

---

### Note · Authors · 2025-12-01

**Comment:**

We sincerely appreciate the time and effort that the reviewers and the program committee have dedicated to evaluating our submission titled “NR-DARTS: Node Rewiring for Differentiable Architectures with Adaptive SE-Fusion”.

After careful consideration, we would like to respectfully request the withdrawal of our paper from the review process. During further internal review, we identified issues that require revisions, and we believe it would not be appropriate to continue the evaluation in its current form.

Thank you again for your understanding and for the invaluable service you provide to the research community. We hope to resubmit improved work to ICLR in the future.

**Withdrawal Confirmation:**

I have read and agree with the venue's withdrawal policy on behalf of myself and my co-authors.